# Photochromic Responses and Stability of Functional Inks Applied on Sustainable Packaging Materials

Sanja Mahović Poljaček * , Maja Strižić Jakovljević * and Tamara Tomašegović

Faculty of Graphic Arts, University of Zagreb, Getaldićeva 2, 10000 Zagreb, Croatia;
tamara.tomasegovic@grf.unizg.hr
* Correspondence: sanja.mahovic.poljacek@grf.unizg.hr (S.M.P.); maja.strizic.jakovljevic@grf.unizg.hr (M.S.J.)

**Abstract:** Photochromism refers to a reversible colour change induced by the irradiation of photochromic materials with ultraviolet (UV) or visible light that reverts to the original colour after the light source is removed. This effect arises from chemical transformations between two isomers with different absorption spectra, involving processes like proton transfer, chemical-bond formation, and isomerisation. These photochromic inks, appearing as crystalline powders with micro-sized particles, require dissolution in a suitable matrix to achieve the colour change. Photochromic inks are used in security, as functional coatings for paper and packaging, in the fabric industry, and in other ways. This study examines the influence of varying concentrations of micro-sized photochromic pigments and different ink-coating thicknesses on the photochromic effect on sustainable paperboard substrates. Artificial ageing was performed to assess the photochromic response and lightfastness in relation to pigment concentration, ink-coating thickness, and the influence of the paperboard substrates. The results of this research could contribute to enhancing knowledge on employing photochromic inks for diverse packaging applications.

**Keywords:** photochromism; functional inks; packaging paperboards; photochromic response; lightfastness

## 1. Introduction

Photochromism is a reversible colour change caused by the irradiation of photochromic materials with electromagnetic radiation. The colour-change effect occurs under the influence of mostly ultraviolet (UV) or visible light in one direction, and in the other direction by fading or removal of the radiation source [1–3]. In photochromism, the colour change is caused by the transformation of chemical bonds between two isomers with different absorption spectra [4,5]. Various mechanisms of photochromic effects, including trans–cis isomerisation, triplet photochromism, and hetrolytic and hemolytic cleavage, cause the colour-change reaction [2,6]. This reversible phenomenon is caused by the removal of the radiation source and the rearrangement of chemical bonds, which cause the molecules to return to their original configuration and colouration.

Photochromic materials belong to the group of chromic materials and can be divided into several types, of which photochromic and thermochromic materials are mostly used [6]. The photochromic materials that change colour when irradiated with UV or visible light and are thermally driven are known as T-type materials, where heat is responsible for the change in colour (thermochromism). If the material returns to its original state only in the absence or fading of irradiation without thermal stimulus, the change in the molecules is photochemical and it belongs to the P-type materials (thermally irreversible but photochemically reversible) [7]. A material that shows colouration and discolouration by UV or visible light changes its state from a thermodynamically stable state A to a metastable state B [4,5]. In general, the stable state A is initially colourless or pale and becomes coloured as it changes to state B under irradiation. With sufficient irradiation or stimulation, switching

between states A and B can be very rapid (duration in the picosecond range) or slow (duration of several minutes to hours) [4].

In most cases, irradiation with UV or visible light is not sufficient to trigger a photochemical reaction in photochromic materials or to achieve the full effect of colour change. Therefore, the irradiation must be of a certain wavelength to be absorbed by the reactant atoms and cause a rearrangement of the bonds between the atoms. Due to the rearrangement of the chemical bonds during irradiation with certain wavelengths, the atoms form a structure that has an intense colour [4,5]. For this reason, photochromic transformation can be controlled by tuning the wavelength of the excitation light, and photochromic materials can be targeted for specific purposes. In addition, photochromic materials are widely used because of their controlled-reversible-photo-response performance, thermal irreversibility, and excellent fatigue resistance [1–3].

Photochromic behaviour is exhibited by organic and inorganic materials and can have a visual response in the visible and invisible parts of the spectrum. For example, UV sensors have been produced based on the principles of photoconductive effects of inorganic photochromic substances [4,8–11]. However, several studies have been directed to organic-based materials rather than inorganic-based materials, because of the greater possibility of achieving a photochromic response in the visible spectrum and of obtaining a colour change visible to the naked eye, which broadens the applications of these materials [6]. In addition, photochromic organic materials are relatively soluble in a dispersive medium and can be easily mixed with or applied to various substrates in the dissolved state. The most used organic photochromic materials are the diarylethenes, fulgides, azobenzenes, spiropyrans, spirooxazines, and naphthopyrans [6,12–15].

In several cases, the photochromic micro-dye is encapsulated in a polymer matrix that provides stability and protects the dye from various substances and chemicals. Furthermore, it has already been published that microencapsulation of photochromic dyes in textiles can improve their lightfastness and increase resistance to light-induced colour change [14,16,17]. This is of great interest for the research of these types of materials, as continued UV irradiation can damage the photochromic dye and alter or block the colour-change effect. Since they can be deposited on different substrates and in combination with different materials, their preparation, application, quality, and colour responses should be carefully studied.

In the packaging industry, photochromic materials can be used for smart labels, as functional coatings or markings, as indicators, and for other purposes. Considering that the use of photochromic effects is extremely interesting, the study of their properties and application is of great importance. To the best of our knowledge, the research results published so far on this topic is modest, especially with respect to the applications on packaging materials that are not standard-paperboard substrates.

Photochromic materials are classified as smart materials and are found in optoelectronic devices and optical storage systems [18–22]; UV markings and sensors; security documents [18]; smart textiles [2,10,23–25]; and elsewhere. They are widely used for their colour-changing properties and have been exploited for attractive designs and for different functional applications. Screen printing, gravure printing, flexographic printing, and offset printing are available methods for the deposition of different inks and coatings with photochromic properties. Inks belonging to the group of photochromic materials usually have the appearance of a crystalline micro-sized powder and, to achieve colour change, must be dissolved in a suitable solvent, which must be selected according to the properties of the substrate and the printing requirements. For this reason, three photochromic inks containing different pigment concentrations were prepared for this research and applied to different packaging substrates. Their colour responses and functional properties were determined as a function of the different paperboard substrates, their concentrations, and of light-induced ageing. The aim of this research was to evaluate the important properties of the photochromic print when the photochromic ink is applied on paperboard substrates that are not conventional, i.e., on sustainable substrates that have been increasingly utilized

in the recent years for various reasons related to ecology and sustainability. If these substrates are to be considered for commercial applications such as value-added packaging, the properties of the prints with special effects on these substrates, including photochromic prints, should be evaluated.

## 2. Materials and Methods

### 2.1. Materials

Photochromic inks were prepared by mixing three types of photochromic pigments with particles in the size range between 2 and 5 µm, based on Potassium Nitrate-Silica-Sodium Nitrate-Azobenzene (Butyl Methoxydibenzoylmethane) (SFXC, Special FX Creative, Life Innovations LTD, Newhaven, UK) in the transparent base used for screen printing (EptaInks, Midrol transparente, Luisago CO, Italy). SFXC® Photochromic Sun Sensitive Reversible Pigments in cobalt blue (CB), orange (OR), and violet (V) were mixed in concentrations of 5%, 7.5%, and 10%. The concentration range of the photochromic pigments was chosen experimentally; the photochromic effect was not vivid if lower concentrations were chosen, and concentrations higher than 10% were not beneficial in terms of improving the visual impression of photochromic effect. The aim was to not use more pigment than needed for cost effectiveness.

The printing process was performed on three types of sustainable paperboard substrates made from alternative resources: Crush Citrus paperboard, with 40% of recycled material and 15% of citrus fruit content ($250 \, g/m^2$); Kaffee Papier, with 5% coffee content ($250 \, g/m^2$); and SH Recycling Grass with 30% grass content ($350 \, g/m^2$) were used. Kaffee Papier and SH Recycling Grass are substrates made completely from recycled materials. The chosen printing substrates belong to the group of environmentally friendly materials produced by replacing a certain amount of virgin tree pulp with process residues of organic products. The use of these alternative materials in papermaking can reduce the potentially negative impact on the environment caused mainly by improper disposal of organic waste. When organic residues are combined with recycled and virgin pulp, the produced paper substrates have a natural appearance, and sometimes tiny particles of waste residues can be visible on the surface. This type of surface texture gives them a strong eco-friendly visual effect that makes them additionally attractive for various designs and applications. They are used for packaging, labels, book covers, promotional materials, greeting cards, etc. [26,27].

### 2.2. Methods

#### 2.2.1. Printing Process

The prepared photochromic inks were screen printed. Screen printing is a printing technique mostly used for applying functional inks to various substrates. It works on the principle of transferring the ink to the substrate using a printing plate with mesh of defined properties. For this study, a printing plate with a mesh density of 32 lines/cm (SEFAR® PET 1500 32/83-100 PW) was prepared for the printing process. The printing plate was made using the conventional method by coating the mesh of the printing plate with photosensitive emulsion [28,29]. The screens were coated three times with AZOCOL Z 133 emulsion from KIWO (Wiesloch, Germany) and exposed to UV radiation for two minutes through the positive films in the Expos-it VASTEX unit, model E2331 (Vastex International, Inc., Bethlehem, PA, USA). During irradiation, the exposed parts of the emulsion were polymerized, and the unexposed parts were rinsed with water. After rinsing, the printing plate was dried in a Dri-Vault drying cabinet (Vastex International, Inc., Bethlehem, PA, USA) and was then ready for printing. Printing was performed with a manual printing machine (Drucktisch 2000 50/70, Bochonow, Birstein, Germany). Printed samples were air dried ($25 \pm 2 \, °C$) for 48 h.

2.2.2. Ageing Process

To determine the influence of the ageing process on the colorimetric properties of printed layers with different photochromic inks, an ageing process was carried out by exposing the samples to xenon radiation in a Solarbox 1500e test chamber (CO.FO.ME.GRA., Milano, Italy). The Solarbox chamber serves to expose the samples to filtered xenon light. It also enables simulation of outdoor weather conditions and provides temperature, rain, and humidity control. The samples were exposed only to xenon radiation, but the relative humidity was not varied (it was kept at $50 \pm 2\%$), since this was not in the scope of this research. An outdoor filter, soda lime glass UV, was used to simulate outdoor irradiation with daylight. Irradiation was set at 550 W/m$^2$ at a standard temperature of 50 °C (in accordance with ISO 4892-2) [30].

Selected samples were subjected to artificial ageing for 1 or 3 h. The short duration of ageing was necessary because the lightfastness of photochromic pigments is generally low and prolonged UV irradiation can damage the photochromic pigment and alter or cancel the colour-change effect.

2.2.3. Characterisation Methods

Images of the surfaces of the paperboard substrates were taken using an Olympus BX51 microscope (Tokyo, Japan) at a magnification of 20×. The thickness (caliper) of the paperboards was measured with the DGTB001 thickness gauge (Enrico Toniolo, Milan, Italy) according to ISO 534:2011. Smoothness was measured with the PTI line Bekk Tester (PTI Austria GmbH, Laakirchen, Austria) on 10 samples for each substrate (5 on the felt side (marked A) and 5 on the wire side (marked B) of the paper) according to the ISO 5627 standard [31]. Optical properties were measured using an X-Rite Exact spectrophotometer, CIE, illuminant D65/10°. The whiteness, yellowness, and brightness of the paperboards were measured over the entire visible range of the spectrum for whiteness and at 457 nm for brightness.

An analytical scale (Mettler Toledo, Columbus, OH, USA) was used to weigh the photochromic pigments and the transparent ink-base. The DLS overhead stirrer (VELP Scientifica Srl, Usmate, Italy) was used to mix the pigments into the transparent base at a constant speed of 50 rpm, for 5 min.

The thickness of the printed inks/coatings was measured using a SaluTron D4-Fe instrument (Frechen, Germany). The SaluTron D4-Fe works on the principle of magnetic induction and measures all non-magnetic surfaces. The ink thickness results were used to determine the possible influence of pigment concentration on the thickness of the printed coatings. ATR-FTIR spectroscopy was used to identify the presence of the functional groups of interest in the surface of the printed photochromic coatings. ATR-FTIR analysis was performed using a Shimadzu IRAffinity-1 FTIR spectrophotometer (Shimadzu Corporation, Kyoto, Japan) with 15 scans per sample. An Ocean Optics USB 2000+ spectrometer (Ocean Optics, Orlando, FL, USA) with a DH-2000 deuterium-tungsten halogen UV light source was used to measure the spectral reflectance and to calculate the colorimetric properties of the printed samples. The Ocean Optics USB 2000+ spectrometer enables spectrometric measurements with a very short integration time (1 ms), which makes it ideal for chemical and biochemical research where it is necessary to observe rapid changes. The range of the measurements is 200–1100 nm, and optical resolution of the spectrometer is ~0.3–10 nm. Measured spectral reflectance of the prints displayed the changes in the photochromic layers' optical properties as an effect of the pigment concentration, artificial ageing, and the influence of the substrate.

## 3. Results and Discussion

### 3.1. Properties of Paperboard Substrates

The microscopic images of the paperboards' surfaces are presented in Figure 1. The observed paperboard samples contain clearly visible fibres intertwined and arranged in different directions in the surface structure. On the surface of Crush Citrus paperboard,

the structure of the fibres is clearly visible, and the remains of the recycled components are visible, given that this substrate consists of 40% recycled fibres (Figure 1a). The surface structure of Kaffe Papier is similar, and both particles and fibres of recycled material are visible in its structure; it consists of processed residues from coffee production and 100% recycled material (Figure 1b). The Recycling Grass shown in Figure 1c is made of recycled paper and contains at least 30% grass dried in the sun. Its surface structure is complex, and it is full of particles of recycled material that are interwoven with different fibres. Microscopic images of all samples reveal tone differences in printing substrates arising from alternative raw materials used in production.

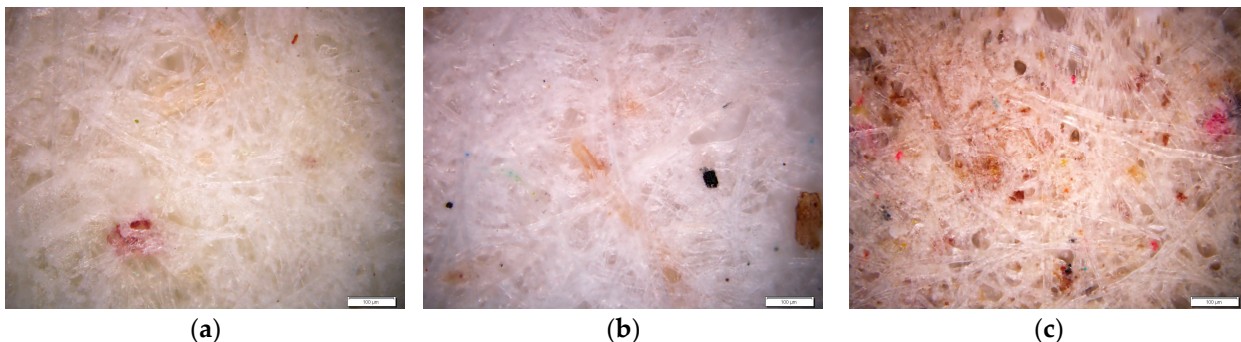

|      (**a**)      |      (**b**)      |      (**c**)      |

**Figure 1.** Microscopic images of the substrates: (**a**) Crush Citrus, (**b**) Kaffee Papier, and (**c**) Recycling Grass (mag. 20×).

The properties of the paperboard substrates are shown in Tables 1 and 2. The preparation of the paperboard samples for the measurements was conducted in accordance with the standard for conditioning paper samples at 23 °C (±1 °C) and 50% (±2%) relative humidity [30]. The results of thickness (caliper), paperboard density, and specific volume (bulk) are presented in Table 1. The results of caliper measurements indicate that the Kaffee Papier sample exhibits the smallest thickness of 0.1707 mm, with Crush Citrus being thicker (0.3237 mm), while Recycling Grass displays the greatest thickness of 0.4845 mm, along with the highest grammage. The grammage and thickness values of each individual sample were utilized to calculate the density of the paper, using the following formula [32]:

$$Y = x/d \cdot 1000 \ (\text{g/cm}^3)$$

where x is basis weight (g/m$^2$) and d is caliper (mm).

**Table 1.** Basic and structural properties of paperboard substrates.

| Paperboards | Caliper (mm) | Basis Weight/Grammage (g/m$^2$) | Density (g/cm$^3$) | Specific Volume/Bulk (cm$^3$/g) |
|---|---|---|---|---|
| Crush Citrus | 0.3237 | 250 | 0.7723 | 1.2948 |
| Kaffee Papier | 0.1707 | 250 | 1.4645 | 0.6828 |
| Recycling Grass | 0.4845 | 350 | 0.7223 | 1.3844 |

**Table 2.** Surface and optical properties of paperboard substrates.

| Paperboards | Smoothness—Bekk Method(s) | | Whiteness (%) W-E-05 | | Yellowness (%) Y-E-05 | | Brightness (%) B-E-05 | |
|---|---|---|---|---|---|---|---|---|
| | A | B | | | | | | |
| | AV | AV | A | B | A | B | A | B |
| Crush Citrus | 6.44 | 5.3 | 31.094 | 28.034 | 20.188 | 20.764 | 68.9 | 67.76 |
| Kaffee Papier | 14.34 | 13.62 | 70.785 | 64.548 | 4.085 | 5.766 | 76.528 | 73.789 |
| Recycling Grass | 1.68 | 1.84 | −13.865 | −23.124 | 27.378 | 28.654 | 41.34 | 36.585 |



The density of a paperboard sheet represents the mass of one cubic centimetre of the tested sample. It is influenced by various additives used in paper production, such as fillers, sizing agents, dyes, fibre type, separation, and mechanical treatments like refining, drying, and calendering. This parameter significantly impacts the paper's optical and mechanical properties, including its structure, porosity, and compactness. Also, paper density is an indicator of the relative air content in the paper [33]. The results presented in Table 1 demonstrate the highest density for the Kaffee Papier sample, with a value of 1.4645 g/cm$^3$, followed by Crush Citrus with 0.7723 g/cm$^3$, and Recycling Grass with 0.7223 g/cm$^3$. This property of paper varies inversely with the specific volume (bulk), which is calculated using the formula [34]:

$$1/Y = d/x \cdot 1000 \ (cm^3/g)$$

where d is caliper (mm) and x is basis weight (g/m$^2$).

The specific volume or bulk is the volume of one gram of the tested paper, cardboard, or corrugated board in space. The results indicate that the Recycling Grass sample has the highest bulk of 1.3844 cm$^3$/g, which also has the highest grammage, followed by Crush Citrus with 1.2948 cm$^3$/g, and Kaffee Papier with 0.6828 cm$^3$/g (Table 1). All samples have high grammage values approaching cardboard, which makes them favourable for screen printing and water-based inks. An additional advantage is their potential application for packaging materials. Residual particles from used raw material in production can be a desirable design element.

Smoothness of the paperboard surface, whiteness (W-E-05), yellowness (Y-E-05) and brightness (B-E-05) are presented in Table 2.

Results of paper smoothness using the Bekk method represent the mean values of five measurements for each individual paper sample tested on both sides, while ten measurements were conducted for all optical properties. Paper smoothness and optical properties significantly influence print reproduction characteristics and quality. The results demonstrate the highest degree of smoothness for the Kaffee Papier sample, with values of 14.34 s for the A side and 13.62 s measured for the B side of the paper. The smoothness of the Citrus Crush paperboard is 6.44 s for the A side and 5.3 s for the B side of the paper, while the smoothness for Recycling Grass is the lowest, with 1.68 s for the A side and 1.84 s for the B side. Although the differences in smoothness between the A and B sides of each individual paper sample are not substantial, they indicate the influence of paper's manufacturing technology on its surface properties. Generally, higher smoothness of the paper results in higher printing quality.

The highest degree of whiteness among the tested samples is exhibited by Kaffee Papier, 70.78% (A) and 64.54% (B); followed by Crush Citrus, 31.09% (A) and 28.03% (B); and finally Recycling Grass, −13.86% (A) and −23.12 (B) (Table 2). Due to its dark shade, Recycling Grass shows negative whiteness values and higher yellowness compared to other samples, which confirms the visual impressions. The value of brightness measured in the narrow spectral region of 457 nm shows the highest values for Kaffee Papier, 76.52% (A) and 73.78% (B); followed by Crush Citrus, 68.9% (A) and 67.76% (B); and Recycling Grass, 41.34% (A) and 36.58% (B). The tested samples indicate that Kaffee Papier, with the highest smoothness, whiteness, and brightness, has the most favourable characteristics for printing.The remaining visible particles of the raw material in paper production (Figure 1b) may be advantageous when choosing this printing substrate for certain purposes and design solutions. The numerical data on measured paper properties can be found in Supplementary materials (Table S1).

### 3.2. Properties of Photochromic Prints

After the inks containing different concentrations of photochromic pigments were prepared, the printing process was carried out and the prints were dried for 48 h. Figure 2 shows images of dried samples printed with 5 and 10% concentrations of the violet pigment taken under ambient light and sunlight, on Crush Citrus, Kaffee Papier, and Recycling Grass substrates. The printed inks are barely visible under ambient light on the observed

substrates (Figure 2a). The inks were prepared by mixing different pigment concentrations (5 and 10%) in a transparent base, followed by printing on different substrates. Ambient light did not trigger the photochemical reaction in the printed coating and a photochromic effect is not visible.

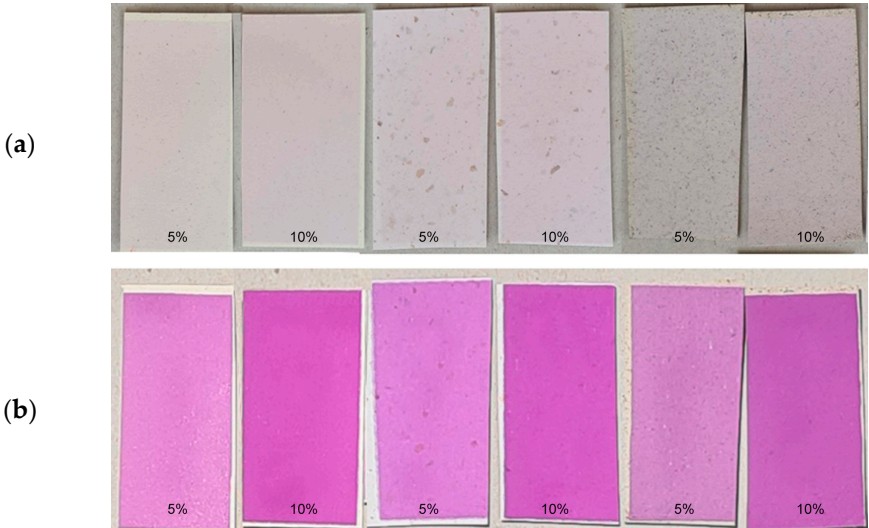

**Figure 2.** Images of the printed coatings printed using different concentrations of the violet pigment (5 and 10%), on, from left to right: Crush Citrus, Kaffee Papier, Recycling Gras. Images were taken in (**a**) ambient light or (**b**) sunlight.

On the other hand, when exposed to sunlight, the influence of the varied pigment concentration is clearly visible (Figure 2b). It was expected that the lowest pigment concentration (5%) would produce the weakest photochromic response and the highest pigment concentration (10%) would produce the strongest photochromic effect. In addition, the difference in photochromic effect caused by different substrates can be visually detected. On the left side of the image, the ink coating on Crush Citrus paperboard can be seen; in the middle, the ink coating on Kaffee Papier; and on the right side, the photochromic ink printed on Recycling Grass is shown. The influence of different substrates is visible on the samples printed with 5% of pigment, where the texture of the paperboards can be observed underneath the print. On the other hand, the samples printed with 10% of pigment show almost completely covering of the surface structure of the paperboards and the texture of the substrates is not visible. Although the pigment concentration is high in these prints (10%), the different properties of the paperboards have had an effect causing differences in photochromic response. It can be seen that the printed coatings on Crush Citrus and Kaffee Papier are brighter than that on Recycling Grass. These results are consistent with the measured whiteness of the studied samples, where Kaffee Papier and Crush Citrus showed higher whiteness compared to Recycling Grass (Table 2). Moreover, these results correspond to the measured brightness values. Kaffee Papier had the highest brightness value, followed by Crush Citrus; the lowest brightness value was detected on the Recycling Grass substrate.

As the inks contained different concentrations of photochromic pigments, the thickness of the printed inks/coatings was measured. The ink thickness results were used to observe the uniformity of the printed coatings and the influence of pigment concentration on the printed coating thickness. Table 3 shows the thickness of printed coatings measured on three paperboards and printed with three different pigment concentrations. The thickness of the printed coatings varies depending on the pigment concentration, regardless of the type of paperboard. The thickness of the photochromic ink layer with 10% pigment in the base is the highest for all samples. It was expected that a higher amount of pigment would result in higher thickness values, but it was not expected that the differences between the

different substrates would be so pronounced. The highest values were found on Recycling Grass and the lowest values on Crush Citrus, regardless of the pigment concentration. Obviously, the different surface structures of the paperboards led to different interactions with the observed printing inks. It is likely that the ink distribution on the paperboard surface was better with Recycling Grass due to it having the lowest smoothness values and the greatest thickness of the substrates. These properties caused better ink transfer in the printing process on Recycling Grass compared to other paperboards.

**Table 3.** Thickness of the printed coatings.

| Paperboards | Conc. Blue Pigment (%) | | | Conc. Orange Pigment (%) | | | Conc. Violet Pigment (%) | | |
|---|---|---|---|---|---|---|---|---|---|
| | 5.0 | 7.50 | 10.0 | 5.0 | 7.50 | 10.0 | 5.0 | 7.50 | 10.0 |
| | Thickness (μm) | | | | | | | | |
| Crush Citrus | 31.5 | 31.5 | 43.9 | 30.5 | 32.5 | 44.1 | 31.2 | 32.4 | 43.9 |
| Kaffee Papier | 34.7 | 36 | 44.3 | 33.9 | 36.1 | 44.6 | 34.2 | 36.2 | 45 |
| Recycling Grass | 51.6 | 49.3 | 70 | 51.7 | 49.4 | 70.2 | 51.3 | 50 | 69.8 |

Figure 3 presents the ATR-FTIR spectra of the photochromic prints with different concentrations of blue, orange, and violet photochromic pigments, respectively. ATR-FTIR spectroscopy was used to detect changes in the printed photochromic ink layer as a result of the varied concentrations of the photochromic pigments.

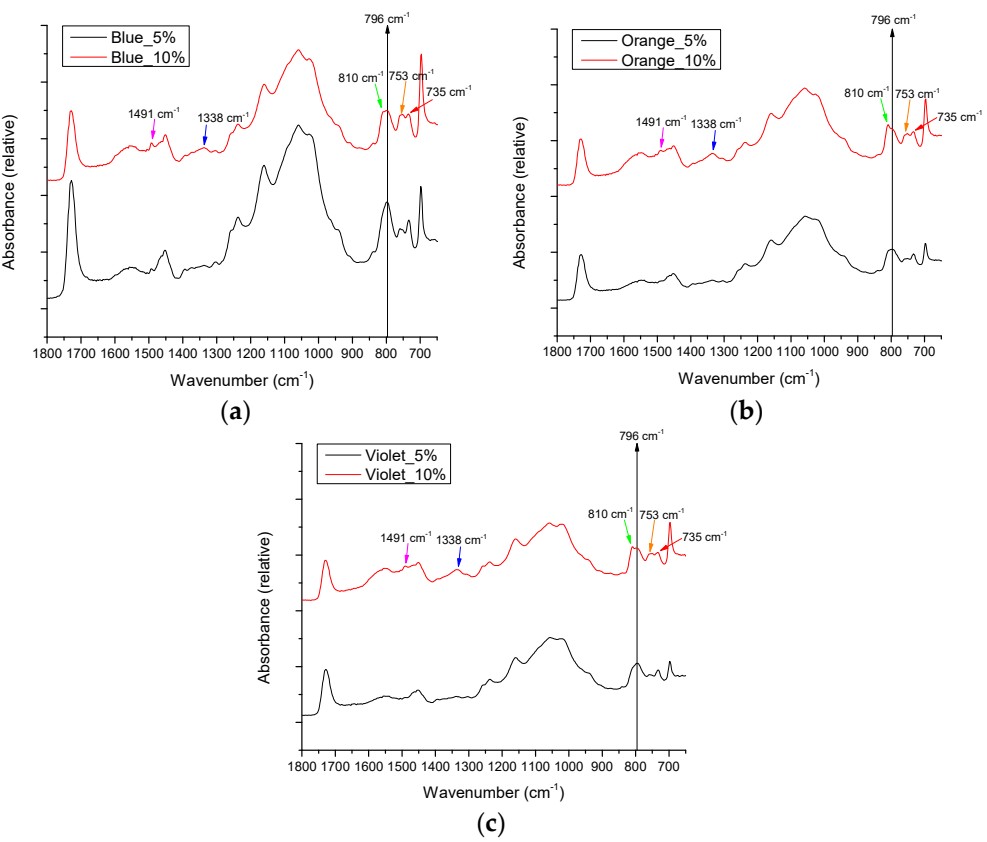

**Figure 3.** ATR-FTIR spectra of photochromic prints with different pigment concentrations: (**a**) blue, (**b**) orange, (**c**) violet.

The ATR-FTIR spectra of the prints with different photochromic pigments are similar, but differences were detected between the prints with 5 versus 10% of the photochromic pigments. The addition of 10% blue, orange, or violet pigment caused changes in ATR-FTIR spectra at $735/753$ cm$^{-1}$, $796/810$ cm$^{-1}$, $1338$ cm$^{-1}$, and $1491$ cm$^{-1}$.

After increasing the concentrations of photochromic pigments from 5 to 10%, the absorbance of the band at 735 cm$^{-1}$ has increased relative to the absorbance of the band at 753 cm$^{-1}$. Furthermore, the increase of the photochromic pigment concentration caused a shift of the band at 796 cm$^{-1}$ to 810 cm$^{-1}$. These bands in the fingerprint region can be assigned to the out-of-plane C-H bending and could point to changes in isomerisation [35]. The changes in the ratios of listed bands can therefore be used only as an indicator of the changes in the ink caused by the varied concentrations of photochromic pigments.

The increased absorbance of the peaks at 1338 cm$^{-1}$ and 1491 cm$^{-1}$ was present for all photochromic prints after increasing the concentration of the photochromic pigments from 5 to 10%. The observed peaks are characteristic for the photochromic pigments, as presented in [36]—specifically, for the photochromic pigments based on azobenzene [37].

The reflectance spectra measured on the printed coatings are shown in Figures 4–6. The reflectance spectra of photochromic inks with different pigment concentrations, printed on different substrates, are presented. Measurements were performed using a spectrophotometer and a DH-2000 deuterium-tungsten halogen UV light source to compare the differences in spectral reflectance of the samples in the visible part of the spectrum.

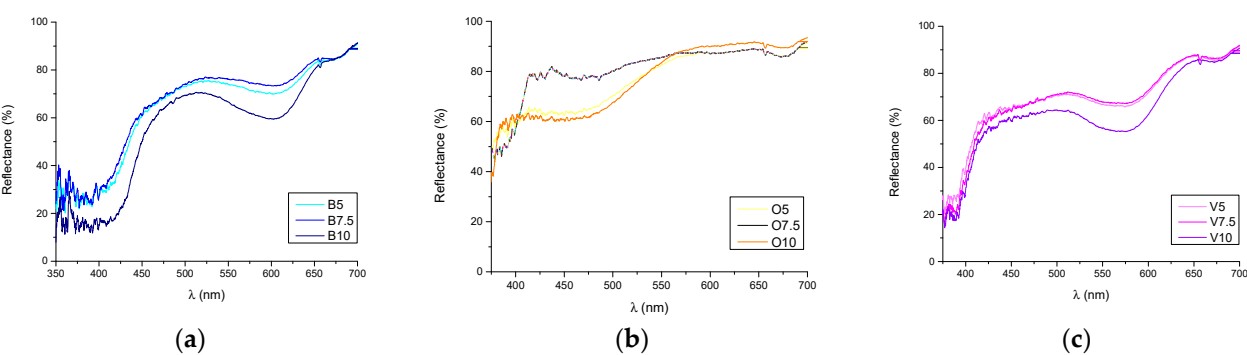

**Figure 4.** Spectral reflectance of inks printed on Crush Citrus with different pigment concentrations: (**a**) Blue, (**b**) Orange, and (**c**) Violet.

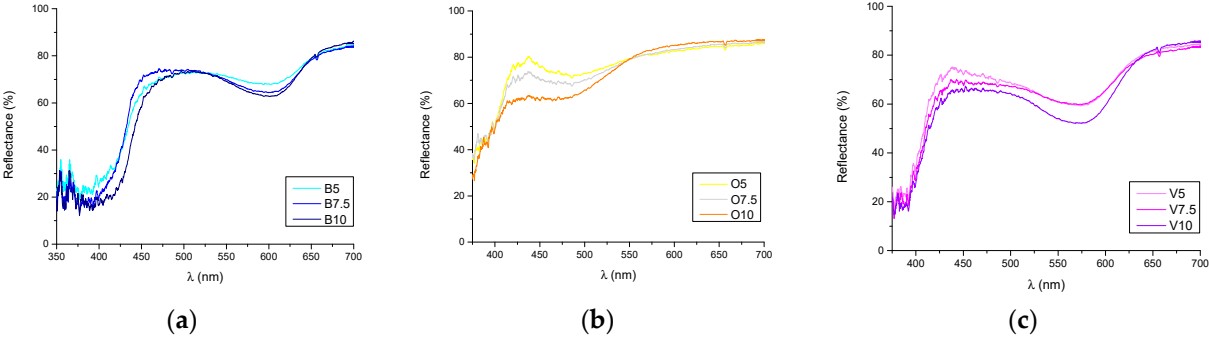

**Figure 5.** Spectral reflectance of inks printed on Kaffee Papier with different pigment concentrations: (**a**) blue, (**b**) orange, and (**c**) violet.

Looking at the overall spectral reflectance results, it can be observed that the concentration of added pigment, and also the printing substrate, has a significant effect on the reflectance curves of the photochromic inks. A higher pigment content (10%) in the ink resulted in better surface coverage of the printed area compared to a lower content (5%). The ink with cobalt blue (B) pigment has the maximum reflectance in the range of 400 to 550 nm, the ink with orange (O) pigment has the maximum reflectance in the range of 560 to 700 nm, and the maximum values of spectral reflectance for inks with different concentrations of violet photochromic pigment (V) were found in the range of 400 to 500 nm. On the other hand, the spectral reflectance of samples printed on the same substrate with different pigment concentrations is quite similar. A pigment concentration

of 7.5% hardly shows any differences compared to the samples printed with a pigment concentration of 5%. Although the difference is clearly visible to the naked eye (Figure 2), spectral reflectance could not clearly detect this difference. One exception is an ink with orange pigment printed on all substrates, where the changes in the spectral reflectance related to the pigment concentration are not clearly visible. The reason for this is probably the radiation source used for the measurement (deuterium-tungsten halogen), which probably did not cause any detectable responses or differences in spectral reflectance.

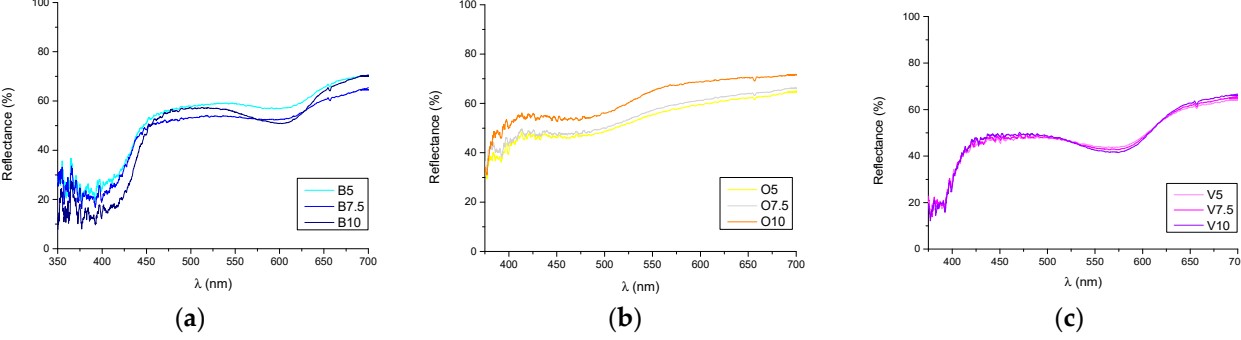

**Figure 6.** Spectral reflectance of inks printed on Recycling Grass with different pigment concentrations: (**a**) blue, (**b**) orange, and (**c**) violet.

### 3.3. Properties of Aged Samples

ATR-FTIR spectroscopy was used to analyse the potential changes in chemical bonds in the surface layers of printing substrates and photochromic prints occurring because of the artificial ageing process.

Figure 7 presents the effect of the ageing on all used printing substrates. Although their compositions were different, the ATR-FTIR spectra of the different substrates are similar, with some minor differences after the ageing process. Specifically, the ATR-FTIR spectra of Crush Citrus paper showed differences in the ratio of the bands at 873 $cm^{-1}$ and 896 $cm^{-1}$, and minor changes in the band around 996 $cm^{-1}$ after the ageing (Figure 7a). The band at 873 $cm^{-1}$ can be assigned to the out-of-plane vibration of carbonate [38] and belongs to $CaCO_3$, which is used as a filler in papers [39].

The band at 896 $cm^{-1}$ indicates the purity of the crystalline band of cellulose [40], while the band around 996 $cm^{-1}$ is characteristic for the fingerprint region of cellulose [41]. The changes in the absorbance ratio of the bands around at 873 $cm^{-1}$ and 896 $cm^{-1}$ after the artificial ageing, coupled with the subtle changes in the band at 996 $cm^{-1}$, point to the changes in the cellulose crystallinity and possible beginning of degradation [42].

Changes in the relative absorbance ratios of bands at 873 $cm^{-1}$ and 896 $cm^{-1}$ after the ageing were also detected in the ATR-FTIR spectra of Kaffee Papier (Figure 7b). However, the ageing process did not affect the Recycling Grass printing substrate (Figure 7c). According to the manufacturer, this printing substrate's surface was pre-treated to improve its printability which could have also caused improved resistance to the ageing.

The ageing process did not cause any visible changes in the ATR-FTIR spectra of the photochromic prints, regardless of the pigment colour and concentration. Therefore, as an example, the comparison of ATR-FTIR spectra of unaged and aged samples are presented only for blue photochromic prints (Figure 8).

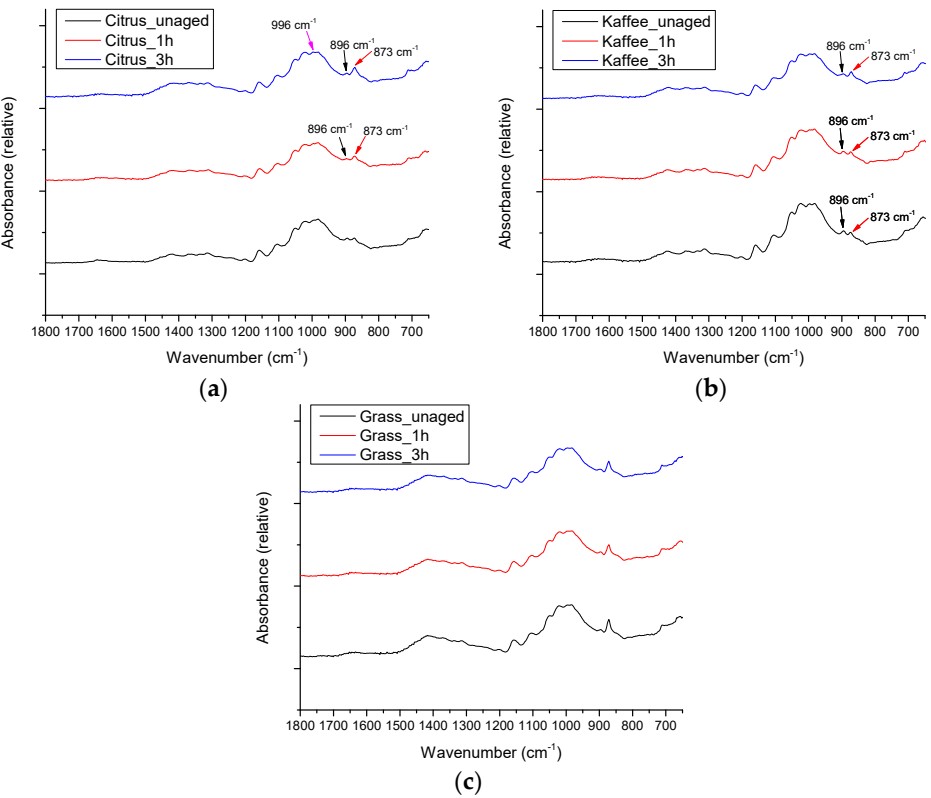

**Figure 7.** ATR-FTIR spectra of unaged and aged substrates: (**a**) Crush Citrus, (**b**) Kaffee Papier, and (**c**) Recycling Grass.

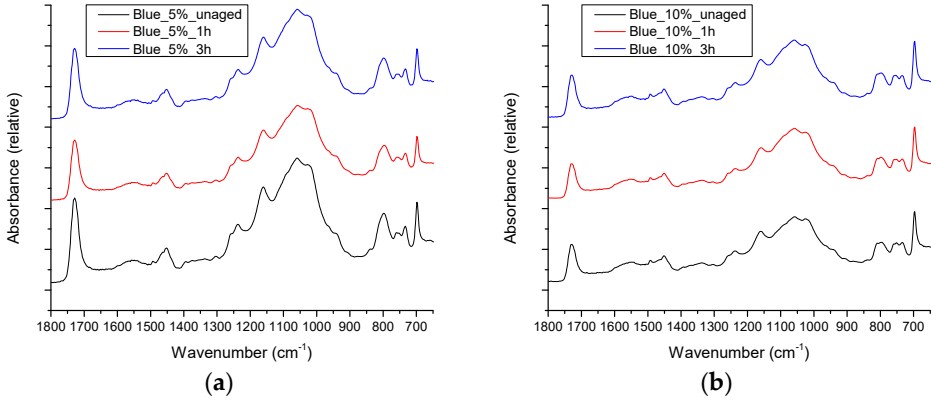

**Figure 8.** ATR-FTIR spectra of blue photochromic prints with different pigment concentrations, before and after ageing: (**a**) 5% and (**b**) 10%.

As discussed in Figure 3, although the ATR-FTIR spectra of the photochromic prints with pigment concentrations of 5 and 10% presented some differences, the ageing process did not cause any visible changes in the ATR-FTIR spectra of prints with the same pigment concentration. Since photochromic prints were visibly affected by the ageing process, the absence of the changes in ATR-FTIR spectra after the ageing indicates that the degradation of the photochromic pigments was not detected by ATR-FTIR spectroscopy, i.e., ATR-FTIR spectroscopy cannot be used as a tool for the assessment of the photochromic pigment degradation. Furthermore, it can be concluded that the transparent base of the photochromic ink did not undergo any degradation after 3 h of artificial ageing.

To define and measure the photochromic response and lightfastness of inks in relation to the concentration of pigments, the printed samples were exposed to artificial ageing for 1 or 3 h. Results of the colorimetric properties of CIE L*a*b* are presented in Figure 9. L*a*b*

system presents three coordinates of the CIELAB colour space; L* represents the lightness of the colour, a* the position between green and red and b* the position between yellow and blue.

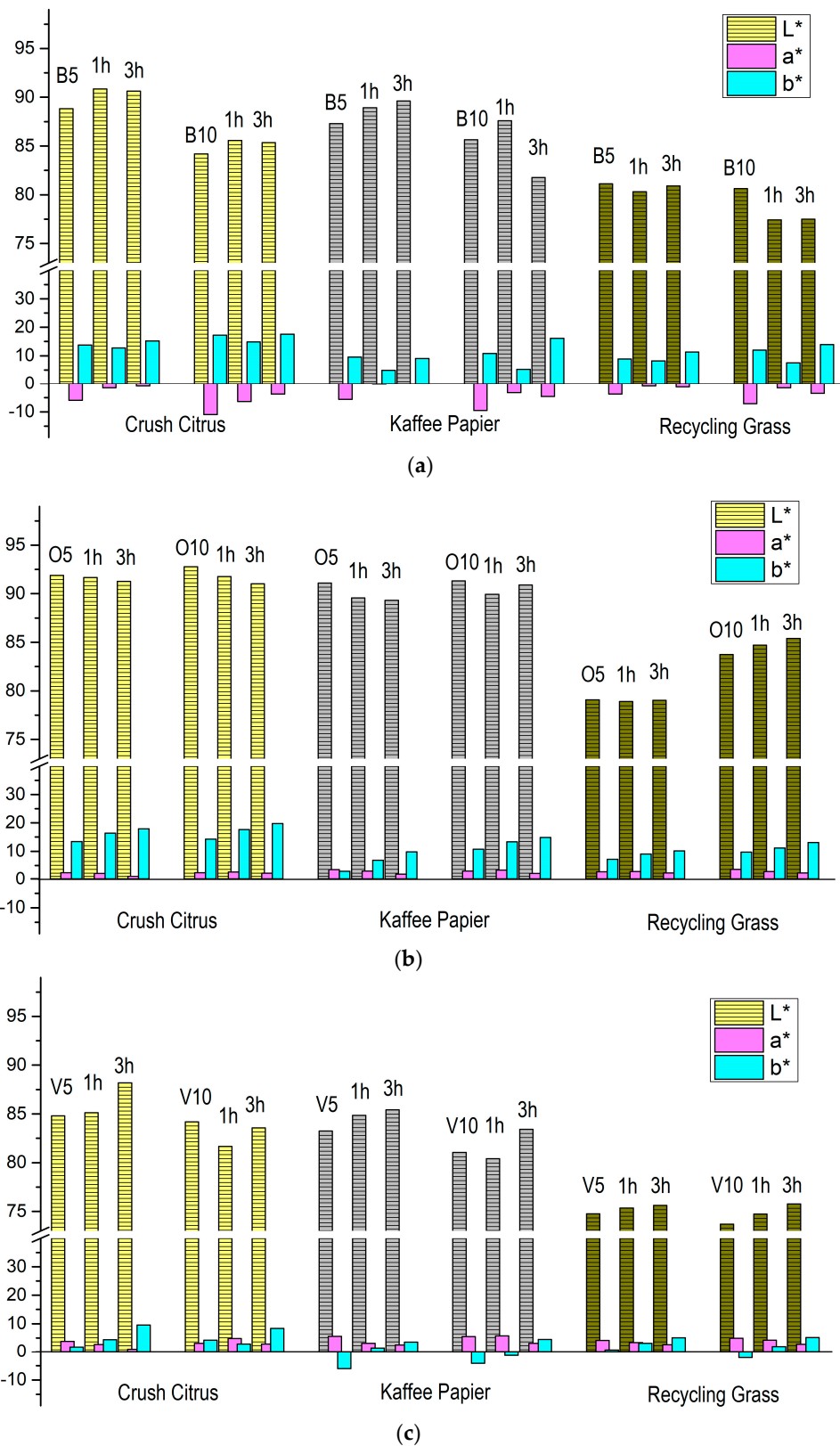

**Figure 9.** Colour parameters in the CIEL*a*b* colour space of photochromic parameters before and after ageing of 1 or 3 h: (**a**) blue, (**b**) orange, and (**c**) violet.

Figure 9a shows the results of L*a*b* coordinates measured on photochromic prints with 5 or 10% blue pigment on all three paperboard substrates. After 1 or 3 h of ageing, all parameters changed compared to the unaged samples. The L* values, which refer to lightness, had increased after 1 and 3 h of ageing. The exception is the L* coordinate measured on Recycling Grass substrate where the L* values were lower. It is visible that all three coordinates had lower values for non-aged samples printed on Recycling Grass. This difference is related to the lightness and brightness of Recycling Grass, mentioned earlier. All unaged samples had a similar a* coordinate, which describes the position of the observed colour between green and red. After ageing, the a* coordinate showed increased values, which means that the position of the observed colour showed a slight transition towards red. The b* coordinate, i.e., the position of the colour between yellow and blue, showed a slight increase after ageing in all samples, which shows the transition of the colour towards yellow.

The results of L*a*b* coordinates measured on photochromic prints with 5 and 10% orange pigment on different paperboards are shown in Figure 9b. It can be seen that the changes in the values are not as pronounced as for prints with blue pigment. The lightness decreased on all prints after artificial ageing. There is no significant difference between the different ageing durations, i.e., the decrease is similar after 1 or 3 h of ageing. The exception was the ink printed on Recycling Grass substrate, for the same reason as described for the blue photochromic ink. Similar values for the a* coordinate were observed for all samples printed on all paperboards. The b* coordinate showed changes after the ageing process. The values increased for all samples after ageing, which means that the photochromic response was directed towards yellow.

Figure 9c shows the results of L*a*b* coordinates measured on photochromic prints with 5 and 10% violet pigment on all observed paperboard substrates. The lightness of the samples on all substrates increased after ageing. The lightness of the unaged samples printed with 10% of pigment was lower than that of the samples printed with 5% pigment. The a* coordinate was similar for all substrates and became lower after the ageing process, which means that the position of the observed colour showed a slight transition towards green. The results of the b*coordinate showed increased values after the ageing process, which means that the measured colour and the photochromic response were directed towards yellow.

The results presented in this research could expand knowledge about the possibilities of photochromic ink application on sustainable paperboard substrates with optical properties that fall outside the standards. Since photochromic inks in graphic technology have found an application in packaging as smart tags or special effects on different types of packaging substrates [43–47], their use could be expanded to prints on ecologically favourable paperboard substrates made of alternative raw materials. This research provided the guidelines for special consideration of the photochromic pigment concentration, the lightfastness of the photochromic prints, and the influence of the printing substrates used on the desired visual effect.

## 4. Conclusions

In this study, the influences of different concentrations of photochromic pigments and different ink-coating thicknesses on the photochromic effect were investigated. In addition, three photochromic inks with different pigment concentrations were prepared and applied to different sustainable packaging substrates. Their colour response and functional properties were determined as a function of the different paperboard substrates, their concentrations, and light-induced ageing. The research results have shown that the properties of the sustainable paperboard substrates used have a significant influence on the photochromic response, because of their optical properties that differ from the properties of conventional paperboards. It was also found that the thickness of the ink layer was related to the pigment concentration in the printed ink layer and the intensity of the photochromic response. The ageing process confirmed that the lightfastness of

photochromic pigments is low and that prolonged UV irradiation, especially over three hours, damaged the photochromic pigment and altered the photochromic response. This observation is important for the planning of the storage and handling of products with photochromic prints. Future research should therefore be focused on the UV-protection of photochromic prints and on enhancing their lightfastness.

Inks with different photochromic pigment concentration have shown some differences in ATR-FTIR spectra. However, regardless of the low lightfastness of the pigments, the ageing process did not cause any visible changes in the ATR-FTIR spectra of the photochromic prints.

The L* values measured on photochromic prints with blue pigment increased after the ageing process. The exception was the L* coordinate measured on Recycling Grass substrate where the L* values decreased. This difference is related to the lightness and brightness of Recycling Grass. The L* values measured on photochromic prints with orange pigment decreased for all prints after ageing (except for prints on Recycling Grass), while the lightness of the samples on all substrates increased after ageing for violet pigment. Shifts of a* and b* values were observed on all prints after artificial ageing.

The application of functional materials that have a photochromic response when irradiated with a specific wavelength can be controlled by tuning the wavelength of the excitation light and can be targeted for specific purposes. Considering that the use of the photochromic effect is extremely interesting, the study of the properties and application of these materials is of great importance. The results of this research have increased knowledge about the use of photochromic inks for various packaging applications, specifically for printing on sustainable paperboard substrates and the effects of a reasonably applied ageing process on the colour shifts of photochromic prints.

**Supplementary Materials:** The following supporting information can be downloaded at: https://www.mdpi.com/article/10.3390/micro4010003/s1, Table S1: Optical properties of papers.

**Author Contributions:** Conceptualisation, S.M.P., M.S.J. and T.T.; methodology, S.M.P., M.S.J. and T.T.; software, S.M.P., M.S.J. and T.T.; validation, S.M.P., M.S.J. and T.T.; formal analysis, S.M.P. and M.S.J.; investigation, S.M.P., M.S.J. and T.T.; resources, S.M.P., M.S.J. and T.T.; data curation, S.M.P., M.S.J. and T.T.; writing—original draft preparation, S.M.P.; writing—review and editing, S.M.P., M.S.J. and T.T.; visualisation, S.M.P., M.S.J. and T.T.; supervision, S.M.P., M.S.J. and T.T.; project administration, S.M.P. and M.S.J.; funding acquisition, S.M.P. and M.S.J. All authors have read and agreed to the published version of the manuscript.

**Funding:** This research received no external funding.

**Data Availability Statement:** Data are contained within the article and supplementary materials.

**Conflicts of Interest:** The authors declare no conflicts of interest.

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
