# Peer review of "Photochromic Responses and Stability of Functional Inks Applied on Sustainable Packaging Materials"

_2673-8023, doi:10.3390/micro4010003_

Round 1

Reviewer 1 Report

Comments and Suggestions for Authors

The authors investigated about the photochromic response and stability of three photochromic inks on sustainable papers. The authors evaluated the photochromic properties from the viewpoint of surface and optical property of sustainable paper, concentration of photochromic inks, and resistance for sunlight. However, for each measurement result, the authors should be discussed more deeply, and it is questionable whether each measurement method is suitable. So, I have decided that this manuscript is not suitable for this journal.

Reviewer 2 Report

Comments and Suggestions for Authors

The authors reported “Photochromic Response and Stability of Functional Inks Applied on Sustainable Packaging Materials”. They focused on the influence of varying concentrations of micro-sized photochromic pigments and different ink coating thicknesses on the photochromic effect on sustainable paperboard substrates. The result shows that the properties of the used sustainable paperboard substrates have a significant influence on the photochromic response. Additionally, the thickness of the ink layer was related to the pigment concentration in the printed ink layer and the intensity of the photochromic response. The principal idea exhibited in this manuscript is good. Thereby, I think the manuscript can be published in Micro after a minor revision. Below are the comments for the authors to address. 

1.     I find a few grammar, spelling and typing mistakes. I highly recommend that the authors should go through the text once again. For instance, the sentence in Line 32-34 needs to be re-written to make it easier to get understood.

2.     All figures need to be improved significantly.

3.     Why were the high concentration (e.g., 15% or 20%) or lower concentration (e.g., 1%, 2%, etc) of the pigment not chosen for this study?

4.     The conclusion session might be extended.

Comments on the Quality of English Language

The quality of English language needs to be enhanced.

Reviewer 3 Report

Comments and Suggestions for Authors

Review Comments:

The article provides a clear and concise introduction to the concept of photochromism and its application in various industries.

The study aims to examine the influence of different factors, such as pigment concentration and ink coating thickness, on the photochromic effect on sustainable paperboard substrates, which is a relevant and interesting topic.

The inclusion of artificial ageing to assess the photochromic response and lightfastness adds an important element to the research, as it provides insights into the durability and longevity of the photochromic inks.

The article emphasizes the potential benefits of using photochromic inks in diverse packaging applications, which highlights the practical relevance of the research.

The research could contribute to enhancing knowledge and understanding of how to effectively employ photochromic inks for specific packaging needs.

It would be helpful if the article provided more details on the specific methods used for the artificial ageing and assessment of photochromic response, as this information would further validate the findings.

Additionally, it would be beneficial to include specific examples or case studies of packaging applications where photochromic inks have been successfully used, to showcase their potential in different industries.

Overall, the article presents valuable research on the influence of pigment concentration, ink coating thickness, and substrate material on the photochromic effect, and its findings contribute to the existing knowledge in the field.

Comments on the Quality of English Language

no

Round 2

Reviewer 1 Report

Comments and Suggestions for Authors

The authors revised the manuscript to emphasize the aim of their research. The reviewer understood the importance of their research. I would like to accept the revised manuscript.